# The immune synapses reveal aberrant functions of CD8 T cells during chronic HIV infection

Nadia Anikeeva [1], Maria Steblyanko[1], Leticia Kuri-Cervantes [2], Marcus Buggert [2,5], Michael R. Betts [2] & Yuri Sykulev [3,4] ✉

Chronic HIV infection causes persistent low-grade inflammation that induces premature aging of the immune system including senescence of memory and effector CD8 T cells. To uncover the reasons of gradually diminished potency of CD8 T cells from people living with HIV, here we expose the T cells to planar lipid bilayers containing ligands for T-cell receptor and a T-cell integrins and analyze the cellular morphology, dynamics of synaptic interface formation and patterns of the cellular degranulation. We find a large fraction of phenotypically naive T cells from chronically infected people are capable to form mature synapse with focused degranulation, a signature of a differentiated T cells. Further, differentiation of aberrant naive T cells may lead to the development of anomalous effector T cells undermining their capacity to control HIV and other pathogens that could be contained otherwise.

Essential role of cytotoxic CD8 T cells in anti-viral immunity is well established[1–4]. Particularly, depleting CD8+ T cells in primates resulted in complete failure to suppress initial burst of SIV replication suggesting that CD8 T cells make critical contribution to control of viremia shortly after infection and during chronic phase of the infection[5,6]. Recognition of infected cells by the T cells result in triggering of T-cell receptor (TCR)- and integrin-mediated signaling that initiates the formation of contact area which transforms into highly organized structure called immunological synapse (IS)[7,8]. The formation of this structure contributes to the coordinated delivery and release of cytolytic granules to target cells, which trigger apoptosis in targeted cells[9,10]. IS at the T cell/target cell interface represents complex 3D structure that is very challenging to study using microscopy because the area is not flat, highly dynamic, and include multiple components, the role of each of them is difficult to ascertain. Glass-supported planar lipid bilayers that mimic the surface of target cells provides unmatched opportunity to analyze T cell contact area, especially dynamics of the molecular events at the interface[10,11]. It has been well established that two proteins, ICAM-1 and pMHC ligands, displayed on lipid bilayer is sufficient to induce the formation of immune synapse by T cells exposed to bilayers[12,13], very similar to that observed at T cell/target cell interface[7,14]. This permits us to evaluate differences in the structure and dynamic of synaptic interface formed by individual polyclonal T cells with various phenotypes and functional activities.

Available data suggest that HIV infection could lead to a global defect in T cells including CD8 T cells[15,16]. Our previous studies of human T cell clones that are derived from peripheral blood of infected people, particularly those with HIV infection, have demonstrated that efficiency of cytolytic activity exercised by cytotoxic T lymphocytes (CTL) is linked to kinetics of TCR-mediated Ca²⁺ signaling and the structure and stability of IS formed by CTL[10,17]. However, whether these synaptic interface parameters identified for T cell clones extend to primary polyclonal T cells of varied stages of differentiation remains unknown. This further solidifies the need to investigating the structure and dynamics of IS of polyclonal T cells from HIV infected and uninfected people using planar lipid bilayers[18,19].

With this in mind, we have examined synaptic interface formation by different subsets of peripheral CD8 T cells from people living with

¹Department of Microbiology and Immunology, Thomas Jefferson University, Philadelphia, PA, USA. ²Department of Microbiology and Institute of Immunology, Perelman School of Medicine, University of Pennsylvania, Philadelphia, PA, USA. ³Departments of Immunology and Medical Oncology, Thomas Jefferson University, Philadelphia, PA, USA. ⁴Sydney Kimmel Cancer Center, Thomas Jefferson University, Philadelphia, PA, USA. ⁵Present address: Department of Medicine Huddinge, Karolinska Institutet, Karolinska University Hospital Huddinge, Stockholm, Sweden. ✉e-mail: Yuri.Sykulev@jefferson.edu

HIV (PLWH) and HIV-negative individuals. We found that CD8 T cells form diverse contact interfaces of different structures and distinct dynamics of accumulation and arrangements of TCR/CD3 and adhesion molecules. The cells with different interfaces exhibited notable dissimilarity in pattern and kinetics of degranulation. The frequency of the cells with distinct synaptic interfaces was dependent on differentiation stage and the infection status. Unexpectedly, we have found that a substantial fraction of naive CD8 T cells from HIV-infected people are able to form mature synapses and release granules. Our findings suggest that chronic inflammation during HIV infection mediates changes in the ability of T-cells to form synaptic interfaces and, consequently, alters their functional activity.

## Results

### The structure of T-cell/bilayer interfaces produced by PBMC-derived CD8 T cells from uninfected and PLWH

We exposed polyclonal CD8 T cells derived from uninfected and HIV-infected people to planar lipid bilayers that display anti-CD3 antibodies and ICAM-1. We analyzed time-dependent changes in the cell adherence to the bilayer surface and variations in the accumulation and segregation of ICAM-1 and anti-CD3 molecules.

The majority of the T cells adhered to the bilayers and established T cell/bilayer interfaces. We examined adhesion area, size, and the dynamic and structure of the interface formation. Most of the cells adhered to the bilayer surface during the initial four minutes after exposure to the bilayers (Supplementary Fig. 1). Cellular adhesion was characterized first by the accumulation and segregation of ICAM-1 molecules and was closely followed by CD3 aggregation. Regardless of donor status, the cells formed different types of contact interfaces, which we categorized into four distinct groups (Fig. 1).

A fraction of cells established a "mature synapse" (Fig. 1a, Supplementary Movies 1 and 2) resembling the classical bulls-eye structure, which was very similar to that observed previously for cloned CD8 T cells[10,13]. The development of this structure began after adhesion arrest and accumulation of ICAM-1 and anti-CD3 molecules at the contact area, followed by the molecular segregation and formation of a mature synapse (Supplementary Figs. 1 and 2). Progression through these stages required up to 20 min, and the vast majority of the cells maintained the structure throughout the 30-min observation period.

A different fraction of cells had a small adhesion area, and wobbled over the bilayer (Fig. 1b, Supplementary Movies 3 and 4). These cells formed "focal synapses" that display accumulated TCR/CD3 molecules without formation of adhesion ring junction (Supplementary Fig. 2). Some of these cells accumulated ICAM-1 at early stage of the interface formation but the accumulation was weak and transient[20] (Supplementary Fig. 1). Compared to mature synapse formation, the kinetics of anti-CD3 accumulation in cells exhibiting focal synaptic structure was somewhat faster, presumably due to the absence of adhesion ring that slows the movement of TCR/CD3 microclusters.

Yet another fraction of T cells formed very unstable dynamic synapses termed "kinapses"[21], displaying migratory morphology with asymmetric adhesion area (Fig. 1c, Supplementary Movies 5 and 6). In most cases, the first phase of kinapse formation resembles that of mature synapses (Supplementary Figs. 1 and 2); however, after adherence and accumulation of ICAM-1 and CD3 molecules the cells lose symmetry and form a leading edge and uropod structure. ICAM-1 molecule accumulated mostly in the middle of the contact area while CD3 concentrated at the uropod.

The last category of T cells had a large adhesion area and peripheral lamellipodium (Fig. 1d). These cells adhered and accumulated ICAM-1 and anti-CD3 molecules with kinetics similar to that observed for mature synapses. However, these CD8 T cells often did not achieve complete segregation of CD3/TCR and LFA-1 during the observation period, and the T cell/bilayer interfaces appeared as multifocal synapses (Fig. 1d, Supplementary Movies 7 and 8).

Changes of the interface structure over time for each category of the T cells are clearly evident from the kymographs demonstrating temporal variations in the spatial position of accumulated TCR/CD3 and LFA-1/ICAM-1 moieties (Fig. 1e–h).

Comparison of adhesion area size and extent of LFA-1/ICAM-1 accumulation for the T cells derived from treated and untreated PLWH or uninfected people revealed statistically significant difference between all T cell categories except those forming kinases or multifocal synapses (Fig. 1i). Extent of ICAM-1 accumulation was similar for each T cell category, with exception of T cells that form focal synapses and do not accumulate ICAM-1 (Supplementary Fig. 3).

Large inter-individual differences in the frequency of T cells that form distinct synaptic interfaces were observed independently of the infection status of donors. However, it has also been shown that humans possess significant inter-individual variations in the frequency of various memory T cell subsets[22]. Thus, the observed differences in the dynamics of synapse formation and the type of interface for polyclonal CD8+ T cells may be linked to distinct differentiation stages that impact the quality of T-cell functioning. With this in mind, we next examined synaptic formation by CD8+ T cell subsets isolated from uninfected individuals and PLWH.

### The ability to accumulate ICAM by CD8+ T cells depends on their differentiation stage and donor status

Using magnetic sorting, we purified CD27+, CD27-CD45RO+ and CD27-CD45RO- CD8+ T cells from PBMC of uninfected individuals and PLWH (Supplementary Fig. 4) and analyzed the ability of T cells to accumulate ICAM-1 at the T cell/bilayer interface followed ICAM-1 engagement by LFA-1 on the T cell surface (Fig. 2). For uninfected individuals, the fraction of T cells capable to accumulate of ICAM-1 was dependent on the stage of T cell differentiation and progressively increased from 40% in CD27+ T cells to 75% observed in CD27-CD45RO-T cells (Fig. 2a). In marked contrast, more than 85% of CD27+ T cells derived from untreated PLWH (Fig. 2b) or treated (Fig. 2c) PLWH accumulated ICAM-1. Furthermore, there was no difference in ICAM-1 accumulation between different subsets of CD8+ T cells from PLWH irrespective of treatment status.

### Comparison of LFA-1 expression on CD8 T cells at various stages of memory differentiation

The accumulation and segregation of the fluorescently labeled ICAM-1 incorporated into the lipid bilayers are mediated by engagement with LFA-1 molecules on T cells. Polyclonal CD8+ T cells, regardless of donor status, contain cells that present with either high or low LFA-1 expression on (Fig. 3a). LFA-1high population expressed about a 3–5-times higher number of LFA-1 molecules per cell compared with LFA-1low populations (Supplementary Fig. 5). Analysis of the LFA-1high population for T cells at various differentiation stages derived from PLWH or uninfected donors is shown in Fig. 3b. In HIV-uninfected individuals, only a small fraction of CD27+ T cells were LFA-1high. In marked contrast, significantly more CD27+ cells were LFA-1high (≈80%) in treated and untreated PLWH. The high level of LFA-1 expression on most CD27+ T cells in PLWH resembles that of T cells at late differentiation stage. The presence of LFA-1high T cells as well as number of LFA-1 molecules per cell in populations of CD27-CD45RO+ and CD27-CD45RO− cells were very similar and independent of HIV infection status (Fig. 3b and Supplementary Fig. 6). ART-treatment did not result in significant change of the fraction of CD27+ cells presenting high level of LFA-1 (Fig. 3b). These data are consistent with the difference in the extent of ICAM-1 accumulation at the interface of CD27 + CD8 T cells from uninfected and infected people (Fig. 2). Together these data suggest that the LFA-1 engagement by ICAM-1 contributes significantly to the differences of the T cells to accumulate ICAM-1 at T cell/bilayers interfaces and to establish mature synapses.

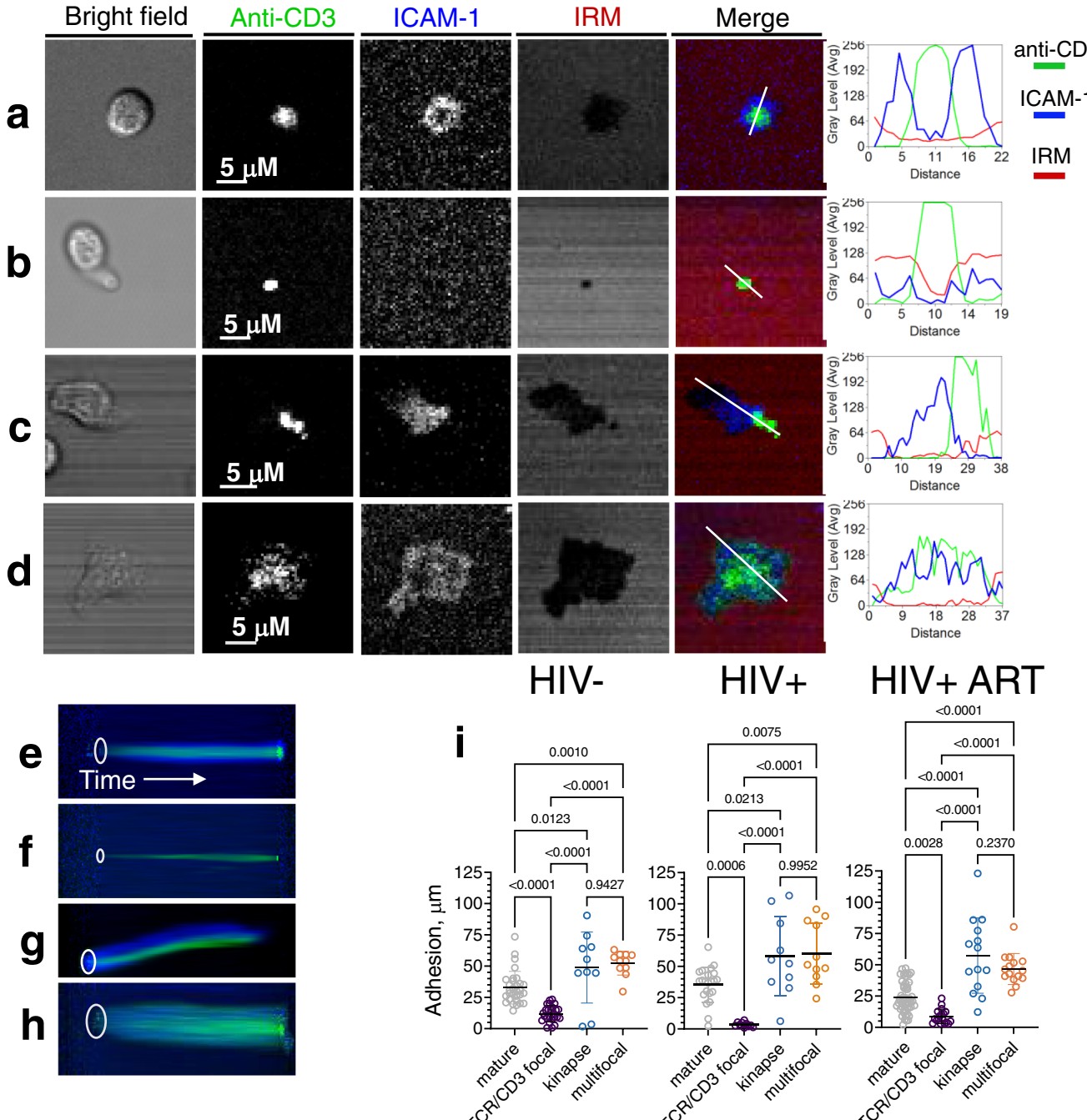

**Fig. 1 | Human polyclonal CD8 T cells establish diverse synaptic interfaces with lipid bilayers that display labeled anti-CD3 antibody and ICAM-1 ligands.** The images were taken by confocal microscope every 2 min over 30 min. Representative images of T cell-bilayer interface taken at the end of the observation period are shown: mature synapse (**a**), TCR/CD3 focal synapse (**b**), kinapse (**c**) and multifocal synapse (**d**). IRM images show contacts of CD8 T cells with the bilayer surface as dark area on light background. Histograms depict the fluorescent intensity profiles along the diagonal white lines in overlay images. Distance is measured in pixels. Blue: ICAM-1, green: anti-CD3, red: IRM. Scale bars: 5 μm. Representative kymographs show temporal changes in spatial position of ICAM-1 (blue) and anti-CD3 antibodies (green): mature synapse (**e**), TCR/CD3 focal interface (**f**), kinapse (**g**) and multifocal synapse (**h**). White oval indicates the position of the cell/bilayer interface at the initial contact of T cell with a bilayer surface, and time axis indicates temporal changes in the spatial position of ICAM-1 and CD3 locations. **i** Variations in adhesion area of CD8 T cells establishing different synaptic interfaces. The adhesion areas were determined from IRM images. For each donor group, the results of representative experiment are shown. The dots on the graphs present individual cells that form mature synapses, TCR/CD3 focal interfaces, kinapses, and multifocal synapses; HIV−: n = 30, 25, 10, 11; HIV+: n = 22, 10, 10, 11; HIV+ ART: n = 52, 18, 14, 15 cells, respectively. Means with SDs are indicated. Adjusted p values were calculated by ordinary one-way ANOVA with Tukey multiply comparison test, and the values are displayed on the top of the graphs. Data (**a**–**i**) represent one out of three independent experiments for HIV- donor group and one out of two independent experiments for each group of infected donors.

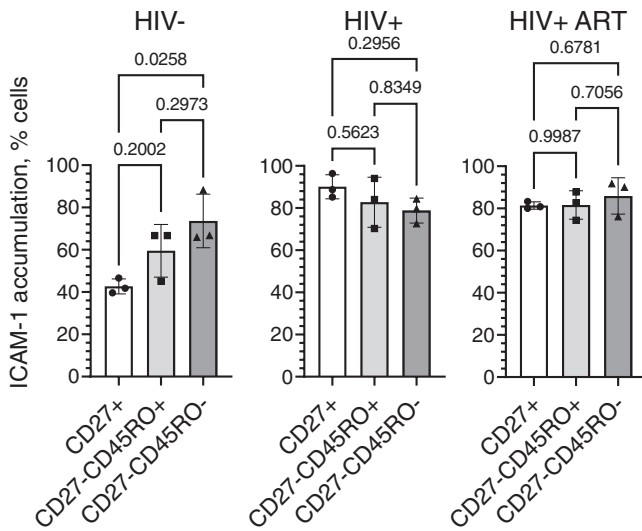

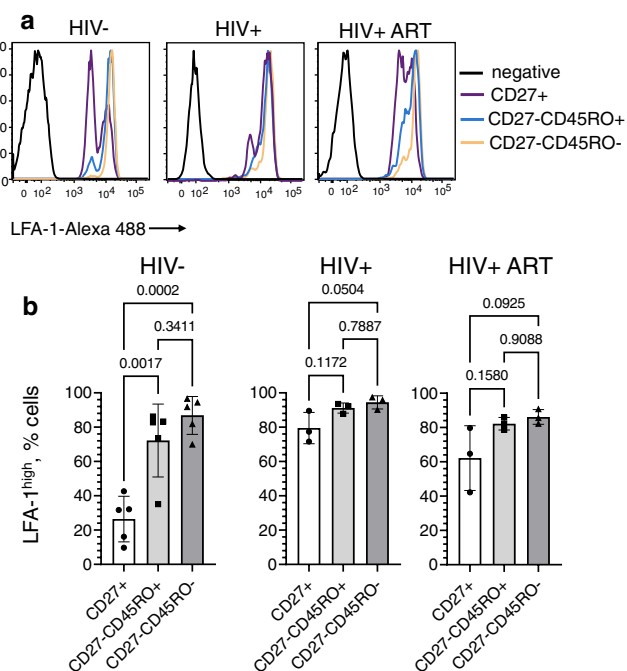

**Fig. 2 | Fraction size of CD8 T cells accumulating ICAM-1 at synaptic interface depends on T cell differentiation stage and donors' status.** CD8 T cells derived from PBMC of HIV− or chronic HIV+ or ART-treated HIV+ individuals were subdivided into CD27+, CD27-CD45RO+ and CD27-CD45RO− subsets by magnetic sorting. Isolated T cell subsets were exposed to bilayers containing labeled anti-CD3 and ICAM-1 molecules, and ICAM-1 accumulation were observed at the interface for 30 min. Bar graphs with error bars indicate mean values with SDs. Each dot point represents independent experiment ($N = 3$ for each donor group). The differences between fraction size within a subject group were determined using ordinary one-way ANOVA with Tukey multiply comparison test; exact adjusted $p$ values indicated on the top of the graphs. Source data are provided as a source data file.

**Fig. 3 | CD27 + CD8 T cells from donors with chronic HIV infection contain significantly higher fraction of cells with upregulated LFA-1 expression, and ART treatment only partially reduce the size of the fraction to a normal level.** The CD8 T cells subsets were isolated by magnetic sorting, stained with anti-LFA antibodies and analyzed by Flow Cytometry. **a** Representative flow histograms showing LFA-1 expression by CD27+, CD27-CD45RO+ and CD27-CD45RO- CD8 T cells derived from HIV−, chronic HIV+, and ART-treated HIV+ donors. **b** Frequency of CD8 T cell subsets with high LFA-1 expression for HIV−, chronic HIV+, and ART-treated HIV+ individuals. For each subset, mean with SD shown as bar graph with error bar. Each dot point represents independent experiment; HIV−: $N = 5$, HIV+: $N = 3$, HIV + ART: $N = 3$. Exact $p$ values calculated by ordinary one-way ANOVA with Tukey multiply comparison test and indicated on the top of the graphs. Source data are provided as a source data file.

## Early differentiated CD8 T cells from HIV+ donors have an enhanced ability to form mature synapses

We next determined the proportions of T cells at various differentiation stages that possessed distinct structures of T cell/bilayer interfaces. The majority of CD27+ T cells derived from PLWH (>60%) formed mature synapses (Fig. 4a, b and Supplementary Fig. 7). In contrast, only a small fraction of CD27+ T cells with this phenotype (~23%) was observed in HIV uninfected donors. Most of the T cells from the HIV− donors formed focal synapses (~70%) characterized by concentrated TCR/CD3 without ICAM-1 accumulation (Fig. 4a, b and Supplementary Fig. 8). A relatively large fraction of CD27+ T cells from ART-treated PLWH established kinapses (~35%). A similar fraction (~32%) of CD27+ T cells from ART-treated PLWH also formed mature synapses; this was somewhat higher compared to T cells derived from uninfected donors (~32% vs 23%) (Fig. 4a, b and Supplementary Fig. 7). In contrast, CD27-CD45RO+ T cells did not show any difference in the ability to establish synaptic interfaces of different kinds between various donor groups. However, CD27-CD45RO− T cells from treated and untreated PLWH established multifocal synapses at significantly higher frequency compared to those from uninfected donors (~30% vs 8%) with a concordant decrease in the fractions of cells that formed mature synapses (Fig. 4a and Supplementary Fig. 9).

## Purified naive CD8 T cells from PLWH exhibit enhanced capacity to form mature synapses

Because CD27 + CD8 T cells contain both naive and memory T cells, it is essential to determine whether purified naive T cells from patients infected with HIV would still have a greater capacity to form mature synapses as compared to naive T cells derived from uninfected donors. To this end, we purified naive T cells as well as transient memory T cells (TM) and effector memory (EM) T cells from HIV-negative and untreated PLWH (Supplementary Fig. 10) and analyzed the formation of synaptic interfaces by each category of these T cells. As evident from

Fig. 5a, naive T cells from untreated PLWH demonstrated an enhanced capacity to form mature synapses as opposed to those T cells derived from uninfected donors. The fractions of T cells that form mature synapses by TM and EM T cells were similar.

Consistent with these results, naive T cells from PLWH revealed an increased ability to accumulate ICAM-1 at the T cell/bilayer interface compared to naive T cells from uninfected donors (Fig. 5b and Supplementary Fig. 11). There was no difference in the extent of ICAM-1 accumulation observed between naive, TM and EM CD8 T cells isolated from HIV infected individuals (Fig. 5b and Supplementary Fig. 11). In contrast, EM T cells from uninfected people revealed a much greater ability to accumulate ICAM-1 compared to naive and TM T cells (Fig. 5b and Supplementary Fig. 11).

These data demonstrate that phenotypically naive CD8+ T cells from PLWH exhibit the behavior of memory T cells.

## Degranulation pattern of CD8 T cells is linked to the type of synaptic interface

Because the amount and pattern of released granules are linked to the efficiency of T-cell effector functions[10,23], we examined the CD8+ T cell degranulation patterns for various types of synaptic interfaces analyzing the appearance of CD107a at the contact cell surface followed by fusion of secretory lysosomes' membrane with the cell membrane (Fig. 6). It has to be noted that the CD107a + endocytic lysosomes do not necessarily carry cytolytic effector molecules but could contain other effector molecules. Granule release in mature synapses was mostly observed in the central supramolecular activation cluster

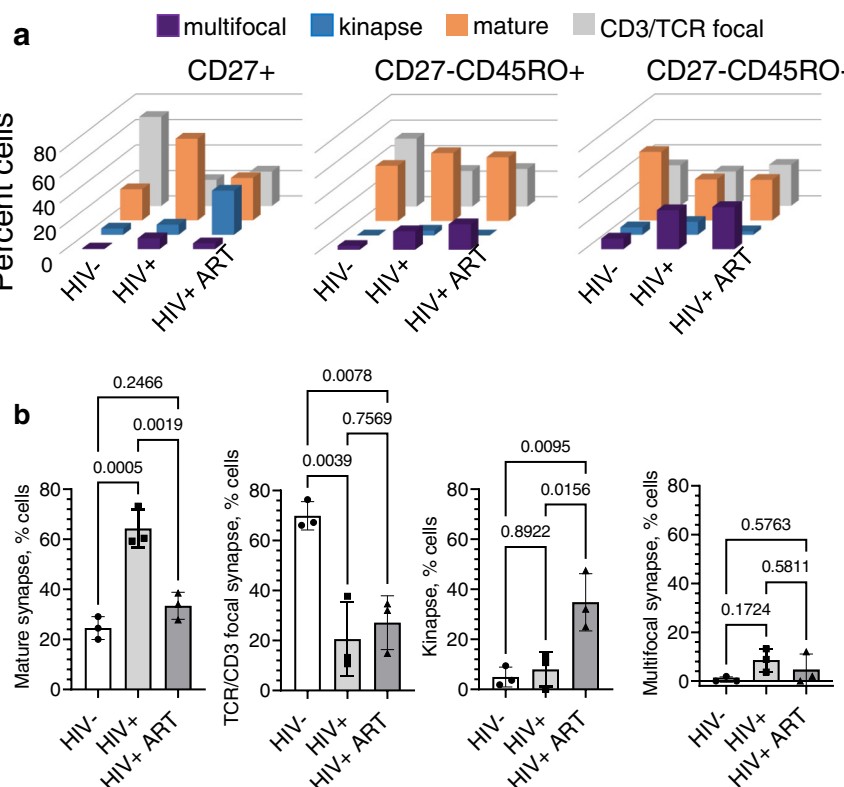

**Fig. 4 | Dissimilarity of the synaptic interfaces established by polyclonal human CD8 T cells from HIV-, chronic HIV+ and ART-treated HIV+ individuals is most pronounce at early stage of differentiation.** CD27+, CD27-CD45RO+ and CD27-CD45RO− subsets of CD8 T cells from HIV-, chronic HIV+ and ART-treated HIV+ individuals were isolated by magnetic sorting and loaded on bilayers containing anti-CD3 antibodies and ICAM-1. The formation of synaptic interfaces was observed for 30 min by confocal microscopy. **a** 3D plots represent frequency of CD27+, CD27-CD45RO+, and CD27-CD45RO− CD8 T subsets that form mature synapses, TCR/D3 focal interfaces, multifocal synapses and kinapses at a T cell-bilayer interface. Infection status of donors are indicated at the bottom of the plots. Mean values from three independent experiments for each donor group are shown. **b** Ability of T cells from HIV−, HIV+, and ART-treated HIV+ individuals to form interfaces of different structures are compared within CD27+CD8 T cell subset. Bar graphs with error bars indicate means values with SDs. Each dot point represents independent experiment (*N* = 3 for each donor group). The difference between subject groups was determined using ordinary one-way ANOVA with Tukey multiply comparison test; exact *p* values indicated on the graphs. Source data are provided as a source data file.

(cSMAC) zone (Fig. 6a and Supplementary Movie 9 and 10) indicative of a short pathway of granule delivery as we have previously shown[10]. The extent of degranulation is steadily increased over time in the cSMAC area (Fig. 6e). T cells establishing focal synapses demonstrated degranulation within a small area around CD3/TCR accumulation, with correspondingly low and diminishing granule release over time (Fig. 6b, f and Supplementary Movie 11 and 12). The weak granule release was associated with transitory ICAM-1 accumulation, without formation of the peripheral supramolecular activation cluster (pSMAC) (Fig. 6f and Supplementary Fig. 1) leading to impermanently less protection of the released granules' cargo from rapid degradation[24]. Granule release in T cells demonstrating kinapse formation was begun at the area of initial bilayer attachment site and continued at the leading edge of moving cells (Fig. 6c, g and Supplementary Movies 13 and 14). These granule delivery patterns and kinetics were associated with diminished functional activity. T cells that form multifocal synapses revealed dispersed granule release patterns over a large adhesion area, indicative of reduced efficiency of released effector molecules (Fig. 6d, h, also see Supplementary Movie 15 and 16). The described patterns of granule release were observed in T cells at all differentiation stages independently of donor infection status but were observed at different proportions.

**Naive CD8+ T cells from PLWH exhibit atypical granule release**
Finally, we examined the granule release characteristics of naive T cells from PLWH. As shown in Fig. 7a, naive T cells from PLWH that formed mature synapses released a larger amount of granules as compared to those cells with TCR/CD3 focal synaptic interfaces. In addition, the released granules within the mature synapses were focused, while the granules appeared randomly over TCR/CD3 focal interface (Fig. 7b and Supplementary Movies 10 and 12). These data suggest that naive T cells from PLWH may exercise some degranulation activity. However, the less differentiated T cells (naive or TM phenotype) from all donor groups showed an overall lower total fluorescent intensity of CD107a staining compared to cells at later differentiated stages (Fig. 7c). T cells at later differentiation stages in PLWH displayed a higher capacity to form multifocal synapse, leading to a scattered pattern of CD107a staining (Supplementary Fig. 12). While no difference in the amounts of released granules was observed between mature synapses and multifocal synapse, the degranulation within multifocal synaptic interfaces demonstrated a dispersed degranulation pattern over large degranulation area (Supplementary Fig. 12a, b, d). In contrast, mature synapses revealed focus degranulation pattern in smaller area, reflective of more potent functional activity (see Supplementary Fig. 12b, c).

## Discussion
Our major findings described here show that majority of naive CD8 T cells from PLWH form mature synapses demonstrating the behavior of activated or memory T cells despite having a naive phenotype. In marked contrast, most naive T cells from uninfected individuals form focal synapses exhibiting aggregation of TCR without aggregation and segregation of LFA-1 (Fig. 5a). This is consistent with the observation that the propensity of naive CD8 T cells from HIV negaitve donors to

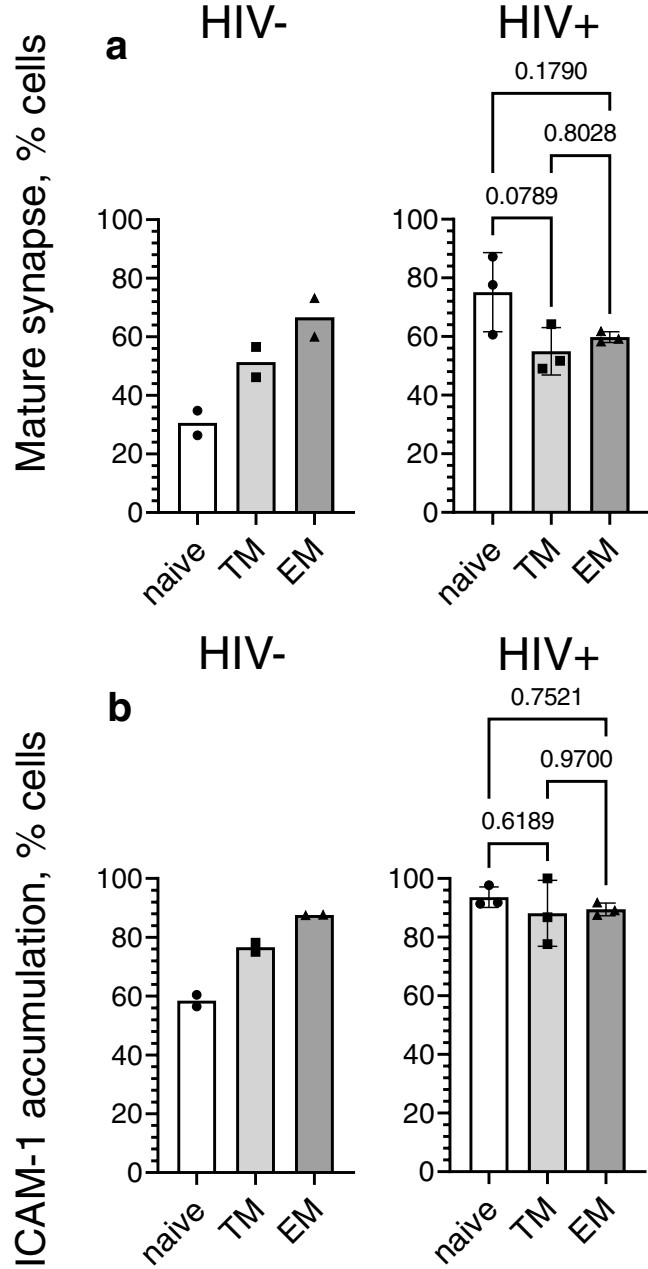

**Fig. 5 | Naive CD8 T cells from untreated HIV+ individuals exhibit greater capacity to accumulate ICAM-1 and form mature synapse compared to T cells from HIV- individuals.** Naive, transitional and effector memory CD8 T cells were exposed to bilayers containing anti-CD3 antibodies and ICAM-1, and formation of mature synapses was observed at 30 min. Frequency of naive, transitional and effector memory T cells establishing mature synapse (**a**) and accumulating ICAM-1 molecules at interface (**b**) are shown. Infection status of individuals are indicated. Two independent experiments were done for HIV− donor group, and data are expressed as average values. Three independent experiments were done for HIV+ subject group and presented as mean with SDs. Each dot point indicate independent experiment. Statistical difference was analyzed by ordinary one-way ANOVA with Tukey multiply comparison test; p values are indicated on the graph (HIV+ donor group). Source data are provided as a source data file.

form synaptic structures is significantly reduced compared to memory T cells[25].

The formation of mature synapses is mediated by TCR-dependent conversion of LFA-1 to a high-avidity form following interaction with ICAM-1 that leads to cytoskeleton remodeling, segregation of highly polymerized actin to the periphery of the T cell contact area, actin clearing from the middle of the interface, and the formation of peripheral ring junction[13,20,24]. The expression of LFA-1 in CD8 T cells depends on the stage of differentiation, with naive T cells presenting the lowest cell surface level of LFA-1 molecules (Fig. 3b). In blood, LFA-1 on naive T cells is presented in low affinity form and could be converted to higher affinity conformation after engaging CCR7 receptors with chemokines during entry of the T cells into lymph nodes[26]. However, signaling through chemokine receptors is not sufficient to induce formation of mature synapses by naive T cells[27], most likely due to inability to form high avidity LFA-1 microclusters. The low affinity form of LFA-1 on naive T cells persists for the first several hours during T cell priming by antigen-presenting dendritic cells resulting in multiple transient contacts of the T cells with APCs allowing preferential activation of T cells with high affinity TCR for the antigen[28].

Similar to the cells from healthy donors, naive CD8 T cells from PLWH presented low level of LFA-1 molecules[29]. However, despite the low level of LFA-1, naive T cells from PLWH have a greater ability to accumulate ICAM-1 and form mature synapses (Fig. 5), perhaps suggesting that chronic inflammation or persistent low level activation during HIV infection[30] could influence productive LFA-1-mediated cytoskeleton remodeling. It has been shown that formation of mature synapses decreases the TCR activation threshold[24]. Thus, an enhanced ability to form mature synapses during first phase of naive CD8 cell priming may result in bystander activation or stimulation of antigen-specific CD8 T cells with lower affinity.

The ability of naive CD8 T cells from PLWH to form mature synapses resembles the behavior of memory T cells (Fig. 5). It has been shown that naive T cell activation preferentially initiates ERK signaling pathways through the SLP76 scaffolding molecule, while in memory T cells, TCR stimulation induces the alternative p38 pathway[31]. Activated ERK acts as negative regulator of LAT/SLP-76 signalosomes facilitating NF-κB signaling and calibrating initial response of naive CD8 T cells to antigens of different strength[32,33]. In contrast, human homolog discs-large protein (hDlg)-dependent phosphorylation of p38 leads to activation of NFAT that is linked to production of cytokines such as IL-2 and IFN-γ[34,35]. The scaffold protein hDlg is recruited to the immunological synapse in response to TCR/CD28 enagagement where it facilitates interaction of Lck with ZAP70 and WASp coordinating actin-mediated synapse formation and effector function[35]. Dlgh1 has been shown to form a dynamic complex with actin-binding protein 4.1 G and LFA-1-binding protein CD226 (DNAM-1) that leads to formation of high avidity LFA-1 clusters required for mature synapse formation. Thus, TCR stimulation on naive T cells from PLWH appears to induce preferentially alternative p38 dependent signaling pathway that links conversion of LFA-1 to the high affinity form, formation of mature synapses and degranulation (Fig. 5).

Until recently, naive T cells have been considered as a 'deeply' quiescent and mostly homogeneous subset[36]. However, recent data have shown that naive T cells are a heterogeneous population, and subset(s) of T cells with a 'classical' naive phenotype can demonstrate transcriptional profiles similar to more differentiated T cells[37,38]. Such naive T cells can also produce various cytokines and are more sensitive to antigen stimulation[38,39]. Unconventional naive CD8 T cells often express chemokine receptor CXCR3, which could direct them to inflamed sites. Presentation of CXCR3 is associated with increased expression of the accessory molecules CD226 which are involved in LFA-1-mediated costimulatory signals for triggering naive T cell differentiation and proliferation[37,38,40–42]. Partial differentiation of naive cells has been attributed to homeostatic proliferation of peripheral naive T cells that arises due to diminished thymic output of naïve quiescent T cells with high T cell receptor excision circle (TREC) content. During untreated HIV infection, homeostatic proliferation of naive CD8 T cells increases three-fold in response to parallel increase of naive CD8 T cell loss[43]. Effective antiretroviral therapy can reverse this loss but does not completely normalize turnover of naive T cells[44].

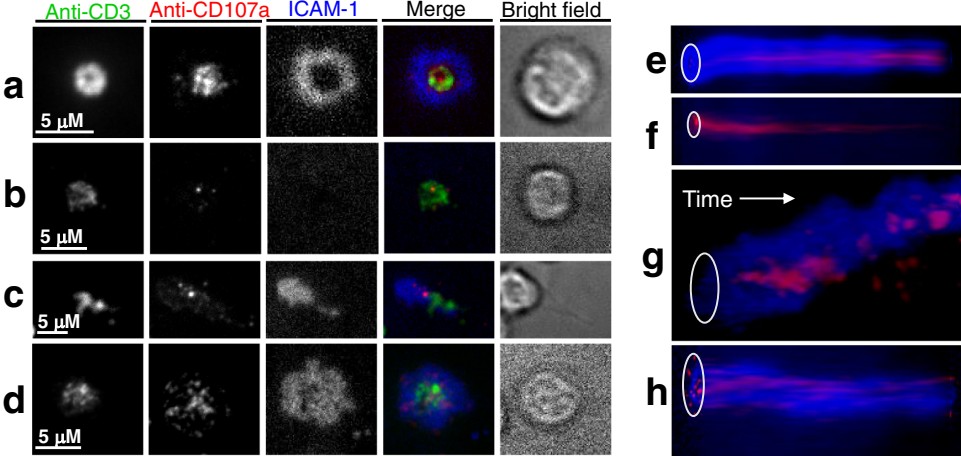

**Fig. 6 | Structure of CD8 T cell-bilayer interface is linked to patterns and kinetics of T cells degranulation.** CD8 T cells were exposed to the bilayers containing anti-CD3 antibodies and ICAM-1 in the presence of anti-CD107a Fab fragments. Structure of the interface and cellular degranulation was observed for half an hour after the initial cell contacts with the bilayers using TIRF (anti-CD3 and anti-CD107a antibodies) and wide field fluorescence (ICAM-1) microscopy. Representative images show relative positioning of degranulation foci (red), andi-CD3 antibodies (green) and ICAM-1 molecules (blue) at the interface of T cells that form **a** mature synapse, **b** TCR/CD3 focal interface, **c** kinapse and **d** multifocal synapse. Images were taken at 30 min after T cell loading on bilayers. Scale bars designate 5 μm. Representative kymographs show temporal changes in spatial positioning of ICAM-1 (blue) and granules (red) released at interface of CD8 T cells forming mature synapse (**e**), TCR/CD3 focal interface (**f**), kinapse (**g**), and multifocal synapse (**h**). Images were taken at the rate of one frame/min for 30 min. White ovals indicate initial position of cell-bilayer interfaces, white arrows show time axis. The images are representative from two independent experiments for HIV− donor group and three independent experiments for HIV+ subject group, *n* = 247, 91, 39, 71 cells forming mature synapses, TCR/CD3 focal interfaces, kinapses, and multifocal synapses, respectively.

Even though the heterogeneity of naive T cells could be mediated by multiple factors including thymic output, chronic inflammation and aging[45], continuous immune activation during HIV infection is a major factor that influence proliferation, graduate loss of stemness properties, and partial differentiation of naive T cells[46,47]. We suggest that these conditions promote the appearance of high avidity LFA-1 on naive T cells that is mediated by alternative p38-dependent signaling pathway, leading to an increase in the ability for synapse formation and degranulation. Indeed, the presence of naive T cells with a higher functional activity in PLWH is evident from their degranulation capacity. In contrast to naive CD8 T cells from HIV- donors, the majority of naive CD8 T cells from PLWH form mature synapses (Fig. 5a) that reveal sustained and focused CD107a staining (Fig. 7b and Supplementary Movie 10) which is similar to memory T cells establishing mature synaptic interface (Supplementary Fig. 12c). However, the observed CD107a staining was still less bright than that of effector memory T cells indicative of a different content of released granules (Fig. 7c)[48,49]. Naive T cells do not produce cytolytic granules, but these T cells may release TNF-alpha or IL-2[38,50]. Cytotoxic granules appear during differentiation from the naive state, and the contents tend to follow a sequential pattern as cells transition towards the effector state, with perforin classically appearing during the effector memory/effector stage[51]. This is not evident for perforin or granzyme B expression by naive cells (even in the case of HIV infection) but given that some degree of degranulation is observed for the naive cells, other granule contents (granzyme A, K, etc.) may be present.

Chronic inflammation also affects plasticity of APC due to the increase of cellular stiffness[52,53]. This enhances their capacity to stimulate T cells changing T-cell metabolic properties and cell cycle progression. At the same time, fluidity of the cell membrane becomes greater as a result of substantial changes in the lipid composition of the membrane[54]. Membrane fluidity that depends on orderliness of membrane lipids facilitates rearrangement of cell surface proteins and the formation of a highly ordered synaptic interface. In addition, IFN-γ induces increase of the expression level of MHC proteins and integrin ligands on antigen-presenting cells that enhances their stimulatory potency.

Transitional memory (TM) cells from both HIV + and HIV- donors were able to form mature synapses and revealed focal degranulation (Figs. 5 and 7b). TM cells maintain tissue trafficking ability and can access lymphoid and non-lymphoid tissues but are thought to be non-cytolytic[55]. The latter is consistent with less dense CD107a staining at the synaptic interface (Fig. 7c). The increased amount of TM cells in HIV+ donors could develop from aberrantly activated naive T cells during HIV infection[56].

Although ART treatment significantly suppresses HIV replication, reduced frequency, and elevated homeostatic proliferation of naive CD8 T cells are still observed[57,58]. Consistent with this, we noticed an enhanced ability to accumulate ICAM-1 at the synaptic interface by early differentiated CD8 T cells from ART-treated PLWH similar to that observed for naive T cells from untreated PLWH (Fig. 2). Even though ART treatment diminished the ability of early differentiated cells to form mature synapses, a significant fraction of these cells still form synapses, albeit with a more dynamic mode (kinapses) as compared to untreated PLWH (Fig. 4b). These kinapse-forming motile cells have a lamellipodium upfront that contains most of the redistributed and accumulated ICAM-1 in a central lamella region followed by a trailing uropod concentrating the engaged TCRs. Degranulation at kinaptic interfaces is not focused, and released granules disperse along the cell trajectories suggesting an impairment in granule delivery. It has been shown that formation of immunological kinapses reflects reduced antigen sensing by T cells[59,60]. Thus, early differentiated CD8 T cells from ART-treated donors are functionally less activated, and a fraction of naive T cells that possess more 'quiescent' properties is increased followed antiretroviral therapy. How treatment duration and age may modulate the synaptic interface structure of naive CD8 T cells in PLWH is unclear and is a topic of future studies.

At late stages of differentiation, significant proportion of T cells from PLWH demonstrated the formation of multifocal synapses with scattered pattern of CD107a staining as opposed to the focused staining typical for CD8 T cells capable to mount efficient cytolytic response (Supplementary Fig. 12)[10,11]. This structure is characterized by a well-developed highly dynamic lamellipodium that persists during the observation period, in contrast to mature synaptic interfaces

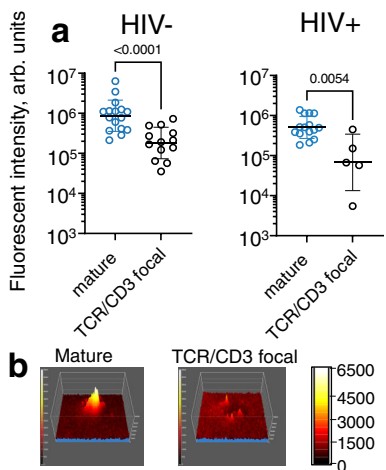

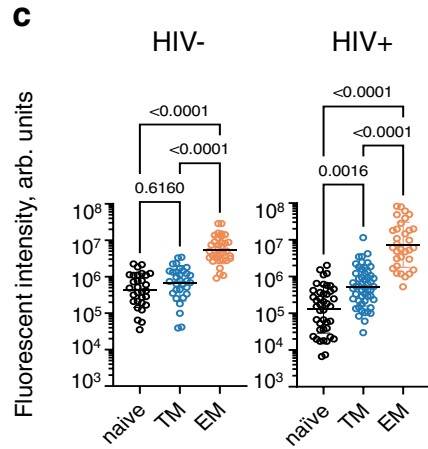

**Fig. 7 | Intensity and quality of degranulation depends on stage of T cell differentiation and type of synaptic interface.** Naive, TM and EM CD8 T cell subsets were isolated by flow cytometry and placed on bilayers containing anti-CD3 antibodies and ICAM-1 in the presence of anti-CD107a Fab fragments. Granule release location and anti-CD3 accumulation were observed by TIRF microscopy. ICAM-1 accumulation was imaged with wide-field microscopy. CD107a accumulation measured at 30 min of initial T cell contact with bilayer surface. **a** CD107a levels measured at mature synapses (blue empty circles) and TCR/CD3 focal synapses (black empty circles) formed by naive CD8 T cells. Infection status of donors indicated at the top of the plot. Each dot represents individual cell; mature: $n = 16$ (HIV−) and $n = 15$ (HIV+), focal: $n = 13$ (HIV−) and $n = 5$ (HIV+). Geometric means with geometrical standard deviation factors shown by black lines. Nonparametric unpaired two-tailed Mann−Whitney test was exploited to determine $p$ values. **b** Representative surface intensity plots showing naive CD8 T cell degranulation at mature and TCR/CD3 focal synapses. The plots are displayed in pseudo-color scale. **c** Quantitation of CD107a accumulation at the interface of naive, TM and EM CD8 T cells from HIV- and untreated HIV+ individuals. Each circle represents individual T cell, HIV−: $n = 30$ (naive), 31 (TM), 34 (EM) and HIV+: $n = 42$ (naive), 52 (TM), 32 (EM) cells. Geometric means are indicated by black lines, error bars represent SD factors. $P$ values are determined by nonparametric unpaired Kruskal−Wallis test with Dunn's test for multiple comparison. **a**–**c** Representative results of one out of three (untreated HIV+ donors) and one out of two (HIV− donors) independent experiments are shown. Source data are provided as a source data file.

where the lamellipodium quickly disappears following initial T cell spreading. This dispersed degranulation pattern can also be observed at synaptic interface formed by NK cells[23,61]. At these conditions, cytolytic granules are released over the synaptic interface at multiple locations devoid of highly polymerized actin[62]. We have previously shown that the formation of stable mature synapses with focused granule delivery increases the efficacy of target cell killing by cytotoxic CD8 T cells by several fold[10,24]. In contrast, our data suggest that terminally differentiated CD8 T cells that form highly motile synapses with scattered granule release have a diminished ability to exercise effector functions. These T cells may represent senescent effector CD8 T cells that are developed in chronically infected PLWH[63,64]. The increased accumulation of dysfunctional CD8+ T cells with signatures of senescent cells, including cell cycle arrest, short telomeres, and diminished of CD28 surface expression, is linked to severity of HIV disease. The abundance of CD8 + CD28- T cells early in the infection is predictive of progression to AIDS[65].

Taken together, our findings demonstrate that previously described CD8+ T cell dysfunctions associated with chronic HIV infection may lead to chronic disturbances in the ability of these cells to properly engage with infected target cells. Importantly, this dysfunction does not appear to be limited only to memory CD8+ T cells but could be also observed in naive CD8+ T cells. In many ways, these changes in CD8+ T cells ability to form synapses in PLWH resemble alterations occurring in the aging T cells. It is likely that the CD8+ T cell defects caused by HIV infection may synergize with similar defects associated with aging further solidifying needs to apply latest techniques for more thorough characterization of these T cells subsets[66,67].

## Methods

### Ethics

Peripheral blood samples were collected from HIV negative donors and ART-treated and untreated PLWH. All samples from PLWH used in this study were de-identified biobanked samples obtained through previously established IRB approved cohorts at the University of Pennsylvania and the University of Toronto. All enrolled participants provided written informed consent conform to Helsinki Declaration, and the protocol was approved by the Institutional Review Board of the University of Pennsylvania (IRB 809316) and the University of Toronto (REB 12-378). HIV + donors did not receive compensation beyond minimal costs associated with travel and time on the donation day. Healthy HIV-negative PBMC samples were obtained from the University of Pennsylvania Human Immunology Core. HIV- donors received $175 for apheresis donation to the University of Pennsylvania Center for AIDS Research, Human Immunology Core. All data specimens were coded to protect confidentiality. Clinical parameters of subjects are summarized in Supplementary Table 1.

### Cells

Hybridoma OKT3 producing antibodies recognizing human CD3ε and hybridoma TS2/4.1.1 secreting antibodies against human LFA-1 were purchased from ATCC (CRL-8001 and HB-244, respectively). Hybridoma H4A3 secreting antibody against CD107a (LAMP-1) was kindly provided by Dr. J. Thomas August, Department of Pharmacology and Molecular Sciences, Johns Hopkins Medical School. H4A3 antibody was deposited to the DSHB by August, J.T./Hildreth, J.E.K. (DSHB Hybridoma Product H4A3). Peripheral blood mononuclear cells (PBMC) were isolated from whole blood using Ficoll-Hypaque density gradient centrifugation and cryopreserved at −140 °C. Hybridoma YN1/1 producing antibodies against ICAM-1 (ATCC, CRL-1878) was kind gift from Michael Dustin, Skirball Institute of Biomolecular Medicine, New York University.

### Antibodies and proteins

Antibody against CD107a was purified from hybridoma H4A3 supernatant. Monovalent Fab fragments of CD107a-specific antibody were produced by papain digestion, purified by anion exchange chromatography and labeled with Alexa Fluor 568. Anti-CD3 antibodies (clone OKT3) were purified from cell culture supernatant. The antibody was mono-biotinylated and labeled with Alexa Fluor 488. Mouse anti-

human antibody against LFA-1 were purified using hybridoma TS2/4.2.1.1 and labeled with Alexa Fluor 488.

Recombinant soluble ICAM-1 protein was expressed in Drosophila Melanogaster cells using pMT/V5-His vector with inducible promotor (Invitrogen), purified by sequential two-step affinity chromatography on anti-ICAM-1 antibody (clone YN1/1) Sepharose and Ni-NTA agarose, and labeled with Cy-5 dye. The affinity Sepharose resin was prepared by coupling purified rat anti-mouse YN1.1 antibody with CNBr-activated Sepharose 4B (Sigma).

The following mouse anti-human antibodies were used for purification of CD8 T cell subsets by Flow Cytometry: anti-CD45RO PE CF594 (clone UCHL1, BD Biosciences), anti-CD56 PE Cy7 (clone B159, BD Biosciences), anti-CD45RA BV650 (clone HI100; BD Biosciences), anti-CCR7 APC Cy7 (clone G043H7, Biolegend), anti-CD27 BV785 (clone O323, Biolegend), anti-CD10 BV605 (clone HI10A, Biolegend), anti-CD14 BV510 (clone M5E2, Biolegend), anti-CD19 PE (clone HIB19, Biolegend), anti-CD16 BV510 (clone 3G8, Biolegend) and anti-CD4 PE Cy7 (clone RPA-T4, Biolegend). The LIVE/DEAD Fixable Aqua Dead Cell Stain Kit (Invitrogen) was used for exclusion of dead cells. Anti-CD3 and anti-CD8 antibodies were excluded from the panel to avoid CD8 T cell stimulation during the staining. After magnetic sorting, purity of CD8 T cells and their subsets were confirmed by Flow cytometry analysis with anti-CD8 Alexa Fluor 488 (clone PRA-T8, BD Biosciences), anti-CD45RO BV650 (clone UCHL1, Biolegend) and anti-CD27 Alexa Fluor 647 (clone O323, Biolegend) antibodies.

All available information about antibodies that have been utilized in this study is summarized in Supplementary Table 2.

## Isolation of CD8 T cells and CD8 T cell subsets
Cryopreserved PBMC were thawed and rested overnight at $2 \times 10^6$ cells/ml in R10 medium (RPMI with 10% fetal bovine serum, 1% penicillin-streptomycin and 1% L-glutamine) with 10 units/ml of DNAse I (Sigma-Aldrich) at 37 °C and 5% $CO_2$. After resting, cells were washed in PBS and used for isolation of CD8 T cells or CD8 T cell subsets by magnetic or flow cytometry sorting.

**Magnetic sorting.** CD8 T cells were purified from PBMC by negative immunomagentic sorting using CD8+ T Cell Isolation Kit (Miltenyi Biotec) and used for bilayers experiments or for further separation into the subsets. CD27 + CD8 T cells were separated with CD27 MicroBeads (Miltenyi Biotec). Negatively selected population were incubated with CD45RO MicroBeads for second round of separation on CD27-CD45RO+ and CD27-CD45RO- CD8 T cell subsets. The cells were washed twice in the assay buffer (20 mM HEPES, pH 7.4, 137 mM NaCl, 2 mM $Na_2HPO_4$, 5 mM D-glucose, 5 mM KCl, 2 mM $MgCl_2$, 1 mM $CaCl_2$, and 1% human serum albumin). After counting the cells were resuspended in the assay buffer at density $2 \times 10^6$/ml and kept on ice prior to experiments. The purity of the subsets (>90%) were confirmed by flow cytometry analysis using anti-CD8, anti-CD45RO, and anti-CD27 antibodies.

**Flow cytometry sorting.** PBMC were pre-stained with antibody against CCR7 in PBS buffer for 10 min at 37 °C. All following incubations were performed at room temperature in the dark. Cells were stained for viability exclusion with LIVE/DEAD Aqua for 10 min. Optimized antibody cocktail was prepared in FACS buffer (0.1% sodium azide and 1% bovine serum albumin in 1X PBS) and was combined with washed cells for 20 min to detect additional surface markers. The cocktail contained mouse anti-human antibodies against CD45RO, CD56, CD45RA, CD10, CCR7, CD27, CD14, CD19, CD16 and CD4. Cells were washed with FACS buffer and resuspended in 350 μl of phenol red-free RPMI. Cells were sorted by BD Biosciences FACS ARIA II with FACSDiva software (v.8.01) at low pressure settings into 1.5 mL DNA LoBind tubes (Eppendorf) pre-filled with 300 μl of R10. The gating strategy is shown in the Supplementary Fig. 8 prepared with FlowJo software (version 9.9.4).

## Measuring LFA-1 expression level
CD8 T cells or purified CD8 T cell subsets were stained with anti-LFA-1 antibody labeled with Alexa Fluor 488 at concentration 2 μg/ml, and the fluorescent intensities were acquired by Flow Cytometry using BD LSRII cytometers with FACSDiva software (v.8.01) and analyzed with FlowJo software (v. 9.9.4). Fluorescent intensity of Alexa Fluor 488 calibration beads with defined number of the fluorescent molecules per bead (Bangs labs Quantum MESF kits) was recorded at the same day and instrument settings. The beads MFI was analyzed with FlowJo software (v. 9.9.4) and plotted against the numbers of the fluorophore per beads using Bangs Laboratory QuickCal Data Analysis Program (v.2.3). The dependence was utilized for conversion of the cell fluorescent intensity values to numbers of the receptors per cell considering that the Alexa Fluor 488/antibody ratio was 4:1 and one antibody molecule could interact with two receptors.

## Planar lipid bilayers
Liposome mixture containing NiNTA-DGS (Avanti Polar) and biotinyl-CAP-phosphoethanolamine lipids at final molar concentration 17.5% and 0.01% correspondingly was used to form bilayer surfaces on glass surface of sticky-Slide VI 0.4 (ibidi) channels. Streptavidin and biotinylated anti-CD3 antibody labeled with Alexa Fluor 488 were sequentially loaded onto bilayers to produce the antibody density of 50 molecules/μm². Cy5-labeled ICAM-1 molecules were introduced into the bilayers through interaction with Ni-NTA-lipids at final density 300 molecules/μm². Channels of ibidi chamber containing bilayers were washed and kept in the assay buffer (20 mM HEPES, pH 7.4, 137 mM NaCl, 2 mM $Na_2HPO_4$, 5 mM D-glucose, 5 mM KCl, 2 mM $MgCl_2$, 1 mM $CaCl_2$, and 1% human serum albumin) prior to use. Detailed protocol for preparation of the lipid bilayers is available[18,19].

## Confocal Imaging
Confocal fluorescent microscopy was carried out on scanning laser Nikon A1R + confocal microscope system equipped with Nikon TiE inverted microscope, Nikon A1plus camera, and four lasers with excitation lines at 405, 488, 561, and 640 nm. The system was fitted with heated stage, objective heater, and a motorized stage with autofocus. The ibidi chamber with preformed bilayers was placed on stage preheated at 37 °C. The cell samples were injected into the entry ports of the slide containing bilayer with displayed ligands. The stage positions were chosen during first two minutes after the sample loading. The selected stage positions were imaged every 2 min for 30 min using 60X/1.4NA objective of the microscope preheated at 37 °C. Brightfield, reflected light, and two fluorescent channels (Alexa Fluor 488 for anti-CD3 imaging and Cy5 for ICAM-1 imaging) of the confocal microscope were utilized to acquire images of interface formed by CD8 T cells interacting with the bilayer surface. The acquisition of the images was performed with Nikon NiS-Elements AR software (v. 4.50.00). MetaMorph (v7.7.2.0) and Nikon NiS-Elements AR (v. 4.50.00) software were employed for analysis and quantification of the images.

## TIRF Imaging
TIRF imaging were employed to assess CD8 T cell degranulation over time. The imaging was performed by Andor Revolution XD system that utilized Nikon TiE microscope with a Nikon TIRF illumination arm (405, 488, 561, and 640 nm laser lines). The system was equipped with 100/1.49NA objective, Andor Ion X3 EM-CCD camera and objective heater. The objective and stage of the microscope were preheated at 37 °C before placement of the chamber with the bilayers on the stage. Alexa Fluor 568 labeled anti-CD107a antibody Fab fragments were added to the cell suspension at a final concentration of 2 μg/mL, and the cells

were injected into the entry ports of an ibidi chamber channel. The stage positions were chosen during first four minutes after cells injection, and the images of the cell-bilayer interface were recorded for 30 min at a rate of one image per minute. TIRF mode were used to assess Alexa Fluor 488 (anti-CD3) and Alexa Fluor 568 (anti-CD107a) fluorescence, widefield was utilized to image Cy5 fluorescence (ICAM-1). The cells morphology was assessed with DIC transmitted light. The images were acquired using MetaMorph Premier Image acquisition software (v. 7.10). Analyses and quantification of the images was performed with MetaMorph Image analysis software (v7.7.2.0 and 7.10).

### Image processing and analysis

To ensure equitable comparison, the measurements of CD8 T cell subsets from individual donors were performed in a series of parallel experiments in the same day using the same reagents and freshly prepared planar bilayers. The images were acquired using the identical microscope settings. Sequential images collected each channel were composed into stacks using MetaMorph image processing software (v. 7.7.2.0). To determine the parameters of cell-bilayer interactions, we chose only CD8 T cells productively interacting with the bilayers. That was determined by accumulation of anti-CD3 antibodies and formation of adhesion area at the interface and confirmed by morphology analysis of the cells observed in the transmitted light images. Clustered cells and visibly damaged or apoptotic cells were excluded from analysis. Cell was discerned to accumulate ICAM-1 if an increase of Cy-5 fluorescence was observed at least on 10 consequent fields. The extent of ICAM-1 accumulation for selected cells was measured by determining the average fluorescence intensity of accumulated Cy5-labeled ICAM-1 molecules at the cell-bilayer interface over background fluorescence outside of the contact area but in close proximity to the cell. If accumulated ICAM-1 molecules formed a ring structure that was observed on at least two consecutive images, we determined that the cell developed pSMAC. The adhesion area corresponds to tight contact between the cells and bilayer and was observed on IRM images as dark region at interface between cells and bilayers. Regions were drawn around tight adhesion areas on the IRM images, and size of those regions were determined.

Productively interacting cells were divided into four categories in accordance with their characteristics of the interface and morphological features. The cell that formed 'classical' pSMAC and cSMAC structures on at least 10 consequent field belongs to mature synapse category. These cells established middle sized adhesion area at the interface and do not form lamellipodia for at least 15 min of observation period. CD8 T cells with sporadic or without ICAM-1 accumulation at bilayer surface classified as cells with TCR/CD3 focal interface. These cells form small adhesion area contacts with the bilayers presumably through uropod-like structure that visible on bright field images in some cases. Motile cells forming asymmetric migratory shape with lamella containing accumulated ICAM-1 and uropod with anti-CD3 antibodies accumulation were defined as cells with kinaptic interface. These cells also characterized by a large asymmetrical adhesion area. The symmetrical cells with large adhesion area and visible lamellipodium structure that observed during almost all period of observation belongs to another category defined as multifocal synapse. Multifocal synapse characterized by multiple small accumulations of anti-CD3 antibodies that can partially coalesce into large central area during period of observation.

### Statistics and reproducibility

Statistical details of the experiments are presented in the figure legends and corresponding Source Data File. The investigators were blinded to perform blood samples collection and allocation to the donor groups. In independent experiments, samples from different donors were tested. No statistical method was used to predetermine sample size. Only data resulting from technical issues such as low quality of a bilayer or poor microscope performance were not utilized for analysis. All data are presented as means with standard deviations with the exception of granule fluorescent intensity data that are presented as geometrical means with geometrical standard deviation factors. Two-tailed $t$- and Mann–Whitney tests were employed to compare two groups. ANOVA, and Kruskal–Wallis tests were used to find difference between three groups. Individual exact $p$-values are indicated on the graphs. Statistical analysis was performed using GraphPad Prism (v. 8.0 and 9.0.1).

### Reporting summary

Further information on research design is available in the Nature Research Reporting Summary linked to this article.

## Data availability

The data for Figs. 1i, 2a–c, 3b, 4b, 5a and b, 7a and c and Supplementary Fig. 1a–c, 3a–c, 5b, 6a–c, 7a–c, 8a–c, 9a–c, 11a and b, 12a and b are available in the Source Data file. Relevant raw microscopy images and flow cytometry data are available from the corresponding author upon reasonable request. Source data are provided with this paper.

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

## Acknowledgements

This work was supported by NIH R01 grant by R01AI11869 to M.R.B and Y.S. M.R.B. is also supported by the Penn Center for AIDS Research (P30-AI045008) and the BEAT-HIV Martin Delaney Collaboratory (UM-1AI164570) which is co-supported by the National Institute of Allergies and Infectious Diseases (NIAID), the National Institute of Mental Health (NIMH), the National Institute of Neurological Disorders and Stroke (NINDS), the National Institute on Drug Abuse (NIDA), and the Robert I. Jacobs Fund of The Philadelphia Foundation. We thank the PLWH who participated in the study and their providers; and M. Ostrowski at the University of Toronto for supplying some samples from PLWH. We are grateful the Sidney Kimmel Cancer Center Bioimaging Shared Resource for excellent support. We also thank the Flow Cytometry Facility of the Sidney Kimmel Cancer Center for excellent technical assistance. We would like to acknowledge the help and support of all members of Yuri Sykulev and Mike Betts laboratories.

## Author contributions

N.A., M.S., M.R.B., and Y.S. conceived of the project, designed, and performed the experiments. N.A., M.S., L.K.-C., and M.B. derived and characterize peripheral blood CD8 T cells from HIV-infected and uninfected people. N.A., M.S., M.R.B., and Y.S. analyzed the data and wrote the manuscript.

## Competing interests

M.R.B. is a consultant for Interius BioTherapeutics. No other conflicts are reported by the authors.

## Additional information

**Correspondence and requests** for materials should be addressed to Yuri Sykulev.

