## [Peer Review File · Nature Communications]

The immune synapses reveal aberrant functions of CD8 T cells during chronic HIV infectionREVIEWER COMMENTS

Reviewer #1 (Remarks to the Author):

The study by Anikeeva et al. NCOMMS-21-46491 is dealing with the fact that HIV-1 infection is causing persistent inflammation, thus premature aging of the T cells, like the senescence of memory and effector CD8 T cells. The authors studied CD8 T cells from chronically HIV-1 infected and non infected people in order to uncover the reasons of gradually diminished potency of CD8 T cells. The authors claim that they found a large fraction of naive T cells developing mature synapses associated with degranulation which is a signature for differentiated T cells. They claim that it leads to the development of anomalous effector T cells in HIV chronic infected people.

The novel finding of this study is about this special population of naive CD8 T cells found in chronically HIV-1 infected people that is showing a mature profile and abnormal degranulation at the synapses. The originality resides in studying CD8 T cell synapse formation on planar lipid bilayer coated with anti-CD3 and anti-ICAM-1. This work is interesting to the field of HIV and immunology.

Authors mentioned their call for functionalized planar lipid bilayer technique because of the limitation of 3D microscopy technique related to the topic which it is not surely an argument. However, the choice of this technique is very good and has been described by the same group in 2015, but the quality of the images could be better (Figure 1). It would have been interesting, for example, to compare CD8 T cell synapses from non infected and in HIV-1 infected cells on APC using 3D Z-stack with high resolution microscopy (like airy scan) to get an insight of 3D cell-cell contacts.

remarks:

Fig1. A-D :

The authors describe qualitatively the observation of 4 categories of T cell/bilayer interfaces. Can authors present some quantitative data here, ie. the % of cells in each category and compare it in HIV- and HIV+ donors.

Fig1. A-D :

In the text, page 4, the authors introduce their question about the 1) adhesion area, 2) dynamic, 3) structure 4) stability of the interfaces. However there's no comment on stability and structure. Where is the link to dynamics ? The result should be explained with respect to the initial questions asked by the authors.

Fig.2 and Fig.3 B:

Data are compared with statistics tests that reveal non significant difference (with big error bar). In this case presenting the data as a scatter plot of all values is recommended.

Page 5, line 13-14 : the sentence seems incomplete.

Fig1. A-D :

The authors mentioned the acquisition of confocal microscope images after 30 minutes of initial cell-bilayered contact. What is the justification of the 30 minutes ? Is this the time required for all the 4 categories to adhere ?

Result 4, page 7 :

Results are again described with qualitative observation despite that numbers are shown in the figure 4.A. Authors should interpretate the results with values.

Fig.5 A, B :

The authors showed that naive cells from HIV-1 infected donor behave as activated cells. Which is strongly related to the new finding in this study. Since authors show previous results with comparison between healthy donor, HIV+ and HIV+ ART donor, it would be interesting to show results (Fig.5) comparing HIV+ and HIV+ART as well.

Result 6, page 8 :

The addition of CD107a is not very well explained or justified in the result section. Results should have a clearer explanation of degranulation and its relation to the figure shown.

Discussion :

The authors discussed cytoskeleton remodeling without any prior reference to it or related experiments involving any actin related proteins or actin labelling. If the discussion is about a required cytoskeleton model then experiments examining differences in synapses and cytoskeletal proteins need to be performed to support these points.

Reviewer #2 (Remarks to the Author):

This is an interesting paper that carefully addresses the interaction modes of CD8 T cell subsets on supported lipid bilayers comparing healthy individuals and HIV infected individuals who are untreated or treated with ART. The authors develop a mice method based on kymographs to classify the interaction with SLB different classes that can be related to earlier efforts. The authors initially divide populations into CD27+ and CD27- and further subdivide the CD27- into CD45R0+ and CD45R0- populations. In healthy individuals most of CD8+ CD27+ cells are naïve thus have lower levels of LFA-1 as expected, but in HIV+ individuals the CD27+ cells may be enriched for additional memory populations and have higher LFA-1. The authors suggest that more precisely defined naïve T sorted based on CCR7 high and CD45RA+ also have a greater tendency to form synapses. The authors look at CD107A surface exposure as a marker of degranulation and conclude that naive T cells from HIV+ individuals also expose more CD107A and that this is more likely to be focused at the center of the synapse.

1. I appreciate that material may not be available to perform new experiments, but request to see some additional data the author should have.

a. It is important to see how the sorting strategy looked for both HIV- and HIV+ in Supplemental Figure 8. The gating looks relatively clean for healthy, but may be a different story for the HIV+. Please show representative data for a HIV+ individual with all panels.

b. The patients used for Figure 1-4 and 5-7 seem to be different. In 1-4 they talk about untreated and ART treated. Then in 5-7 they talk about "chronic". Are we to assume these are untreated? Can the three patients used in 5-7 be broken out separately in Table S1 to make this clearer.

c. Since LFA-1 analysis is not provided for the sorted cells the authors should provide single cell data for ICAM-1 accumulation of the naïve, T_{hm} and T_{em} subsets with ratio or contact to surrounding bilayer pooled for all categories. This should help to understand if the naïve HIV+ T cells have higher LFA-1 levels, even though LFA-1 expression is not the only factor.

2. The authors use CD107A as a surrogate for degranulation, but what does this mean for naïve T cells? Do the naïve T cell subset in the HIV infected individuals express perforin, granzymes or FAS? Did the authors save the supernatants so they could do a serine esterase release assay? It would helpful to have any insight into who the authors see the functionality of CD107A expression in that naïve subset and if there is an “armed” naïve population in the chronically infected individuals.

Below are point-by-point answers to reviewers' questions.

Reviewer #1

It would have been interesting, for example, to compare CD8 T cells synapses from non infected and in HIV-1 infected cells on APC using 3D Z-stack with high resolution microscopy (like airy scan) to get an insight of 3D cell-cell contacts.

Answer: While it is interesting to study the detailed 3D structures of T cell-APC conjugates with polyclonal T cell subsets from different donors, quantitative comparison requires multicolor time-lapse imaging and analyses of several fields with dozens of randomly positioned unsynchronized conjugates for each sample. Currently, this is a non-trivial task for live time-lapse super resolution microscopy (Front Immunol., 2018, 9:684). As our strategy using lipid bilayers does not require the use of specific target cells, developing a two cells assay would be a substantial level of development that would be more challenging than with T cell clones and APC cell lines. In addition, a redirected T-cell system would need to be developed and optimized to study polyclonal CD8 T cells of unknown specificity. Thus, an independent project would have to be executed to develop the strategy.

1. Fig1. A-D:

The authors describe qualitatively the observation of 4 categories of T cell/bilayer interfaces. Can authors present some quantitative data here, i.e., the % of cells in each category and compare it in HIV- and HIV+ donors.

In the text, page 4, the authors introduce their question about the 1) adhesion area, 2) dynamic, 3) structure 4) stability of the interfaces. However, there's no comment on stability and structure. Where is the link to dynamics? The result should be explained with respect to the initial questions asked by the authors.

The authors mentioned the acquisition of confocal microscope images after 30 minutes of initial cell-bilayer contact. What is the justification of the 30 minutes? Is this the time required for all the 4 categories to adhere?

Answer:

When we tested unsorted total population of CD8 T cells, we observed a large interindividual variability in the response to stimulation with antigen-presenting bilayer surface. As human donors possess significant inter-individual differences in the frequency of various T cell subsets (Nature Medicine, 2019, 25:487–495), we decided to isolate and analyze different CD8 T cell subsets. Indeed, we found that the percent of T cells of each category varies and depends on the stage of T-cell differentiation and donors' status, i.e., HIV-infected or uninfected or ART treated. These data are presented in Figure 4.

To evaluate the dynamics of T cell-bilayer interactions, we prepared a new supplemental figure (Suppl Figure 1). For each kind of interface, we now present graphs showing time-dependent changes in the frequency of cells adhered to the bilayer surface, frequency of cells accumulating ICAM-1, and the frequency of cells establishing specific synaptic interface type. To quantify differences in the structure of various interfaces, we built intensity profiles across images of cells that established different kind of interfaces (Suppl Figure 2).

We also applied extended definitions of varied immune synapse structures, including the true immunological synapse (defined as a central cluster of TCR molecules surrounded by a ring of adhesion molecules) as well as non-classical structures including kinapses and multifocal synapses.

Since the dynamics and stability of the synapse are overlapping terms, we removed word "stability".

Finally, we chose 30 minutes intervals from the dynamics of synapses formation (see Supplemental Figure 1). The vast majority of the cells adhered during first 10 minutes, while the formation of the interface is completed within 20 minutes.

2. Fig.2 and Fig.3 B:

Data are compared with statistics tests that reveal non-significant difference (with big error bar). In this case presenting the data as a scatter plot of all values is recommended.

Answer:

We will present the data shown in Figure 2 and Figure 3B as scatter plots.

3. Result 4, page 7:

Results are again described with qualitative observation despite that numbers are shown in the figure 4.A. Authors should interpretate the results with values.

Answer:

We have now incorporated the average percentage of each category of T cells derived from HIV+, HIV- and ART-treated donors that form the indicated structures of synaptic interfaces into the text on Page 7.

4. Fig.5 A, B:

The authors showed that naïve cells from HIV-1 infected donor behave as activated cells. Which is strongly related to the new finding in this study. Since authors show previous results with comparison between healthy donor, HIV+ and HIV+ ART donor, it would be interesting to show results (Fig.5) comparing HIV+ and HIV+ART as well.

Answer:

The major finding of this study is a striking difference in the ability naïve T cells from HIV-1 infected and uninfected people to establish mature synapses. While this was very clear from analysis of the T cell/bilayer interface of CD27+ T cells representing mostly naïve T cells, we had to isolate CD45RA+ CCR7+ cells to eliminate a small fraction of central memory T cells that are present in CD27+ population. We agree that it would be interesting to systematically analyze HIV+ ART donors and to determine how the behavior and functions of T cells depends on the time of ART initiation, length of ART treatment, and type of antiviral drug cocktails. This study would have to include additional tests since the effect of treatment may be very different for each patient. Based upon the need for extended studies to conduct a careful comparison of HIV+ ART treated patients, we feel that this is beyond the scope of this project. We can address this issue in the discussion if required.

5. Result 6, page 8:

The addition of CD107a is not very well explained or justified in the result section. Results should have a clearer explanation of degranulation and its relation to the figure shown.

Answer:

CD107a is a marker of degranulation, an essential function of cytotoxic CD8 T cells. The pattern of granule release determines efficiency of granule content delivery to target cells and is readily measured by CD107a deposition at the immunological synapse. In the Results section we emphasized that the structure of synaptic interface is linked to pattern (and therefore efficiency) of granule release.

6. Page 5, line 13-14:

the sentence seems uncomplete.

Answer:

The sentence has been corrected

7. Discussion:

The authors discussed cytoskeleton remodeling without any prior reference to it or related experiments involving any actin related proteins or actin labelling. If the discussion is about a required cytoskeleton model then experiments examining differences in synapses and cytoskeletal proteins need to be performed to support these points.

Answer:

Thorough analysis of cytoskeleton remodeling would require executing a separate project, and the samples from donors used for the described experiments are no longer available. We did incorporate references regarding cytoskeleton remodeling.

However, if the reviewer would like to recommend quoting other references that we missed and are more appropriate, it would be very helpful.

Reviewer #2

1. I appreciate that material may not be available to perform new experiments, but request to see some additional data the author should have.

a. It is important to see how the sorting strategy looked for both HIV- and HIV+ in Supplemental Figure 8. The gating looks relatively clean for healthy but may be a different story for the HIV+. Please show representative data for a HIV+ individual with all panels.

b. The patients used for Figure 1-4 and 5-7 seem to be different. In 1-4 they talk about untreated and ART treated. Then in 5-7 they talk about "chronic". Are we to assume these are untreated? Can the three patients used in 5-7 be broken out separately in Table S1 to make this clearer.

c. Since LFA-1 analysis is not provided for the sorted cells the authors should provide single cell data for ICAM-1 accumulation of the naïve, T_{tm} and T_{em} subsets with ratio or contact to surrounding bilayer pooled for all categories. This should help to understand if the naïve HIV+ T cells have higher LFA-1 levels, even though LFA-1 expression is not the only factor.

Answer:

a. We now include the requested gating analysis of HIV+ donor samples in Supplemental Figure 8 as requested.

b. Yes, 'chronic' referred to untreated HIV+ infected donors. We have corrected the inconsistency in terminology. Description of HIV-infected donors whose T cell subsets were isolated by flow sorting for analysis is provided in Supplemental Figure 10.

c. We have now determined the pooled ratios of ICAM-1 accumulation at single cell level for naïve, T_{tm} and T_{em} subsets. These data presented in new Supplemental Figure 11 and described in the Results section.

2. The authors use CD107A as a surrogate for degranulation, but what does this mean for naïve T cells? Do the naïve T cell subset in the HIV infected individuals express perforin, granzymes or FAS? Did the authors save the supernatants so they could do a serine esterase release assay? It would helpful to have any insight into who the authors see the functionality of CD107A expression in that naïve subset and if there is an "armed" naïve population in the chronically infected individuals.

Answer:

Naïve T cells do not produce cytolytic granules (a specialized lysosome), but these T cells may release TNF-alpha or IL-2 (J Immunol, 2005, 175: 5043; doi J Immunol, 2019, 203:3179). Cytotoxic granules appear during differentiation from the naïve state, and the contents tend to follow a sequential pattern as cells transition towards the effector state, with perforin classically appearing during the effector memory/effector stage (J Leukoc Biol. 2009, 85:88). There indeed is not evidence of perforin or granzyme B expression by naïve cells (even in the case of HIV infection) but given that some degree of degranulation is observed for the naïve cells, other granule contents (granzyme A, K, etc) may be present. In the future, we are planning to execute project focusing on analysis of functional activity and gene expression pattern of “shallow” quiescent naïve T cells from HIV infected donors, but it would take significant efforts and time.

REVIEWERS' COMMENTS

Reviewer #1 (Remarks to the Author):

The manuscript is now acceptable for publication as the authors answered the majority of the requests.

Minor:

- define PLWH in the introduction.
- keywords need to be reduced as keywords (it should not be sentences)

Reviewer #2 (Remarks to the Author):

The authors have addressed my concerns. The new data provided in response to my comments and those of the other reviewer also improve the paper in that it provides all the information needed for a reader to appreciate the quantitative aspects of the data with regard to classification of different interaction modes and kinetics of specific features. It is important that the naive T cells in the HIV setting express levels of LFA-1 that are more similar to a memory cell, which may explain some of the change in synapse formation.

Response to reviewers' comments

We are grateful to reviewers' comments on the revised manuscript and their suggestions. We made appropriate corrections to the text as has been suggested. All changes in

Reviewer #1 (Remarks to the Author):

The manuscript is now acceptable for publication as the authors answered the majority of the requests.

Minor critiques:

- define PLWH in the introduction.

Answer: PLW, i.e., People Living With HIV, has been spelled out in the Introduction

- keywords need to be reduced as keywords (it should not be sentences)

Answer: We have significantly reduced key words: HIV-infection, CD8 T cells, stages of differentiation, lipid bilayers, immune synapses, degranulation patterns

Reviewer #2 (Remarks to the Author):

The authors have addressed my concerns. The new data provided in response to my comments and those of the other reviewer also improve the paper in that it provides all the information needed for a reader to appreciate the quantitative aspects of the data with regard to classification of different interaction modes and kinetics of specific features. It is important that the naive T cells in the HIV setting express levels of LFA-1 that are more similar to a memory cell, which may explain some of the change in synapse formation.

Answer: The reviewer provided just-on-target summary of our findings